# Comparative Evaluation of Band-Based Genotyping Methods for *Mycobacterium intracellulare* and Its Application for Epidemiological Analysis

**DOI:** 10.3390/microorganisms8091315

**Published:** 2020-08-28

**Authors:** Jeong-Ih Shin, Jong-Hun Ha, Dong-Hae Lee, Jeong-Gyu Choi, Kyu-Min Kim, Seung Jun Lee, Yi Yeong Jeong, Jong Deog Lee, Myunghwan Jung, Seung-Chul Baik, Woo Kon Lee, Hyung-Lyun Kang, Min-Kyoung Shin, Jung-Wan Yoo

**Affiliations:** 1Department of Microbiology, College of Medicine, Gyeongsang National University, Jinju 52727, Korea; jung20787@gmail.com (J.-I.S.); hjh8824@gmail.com (J.-H.H.); hanwa5@nate.com (D.-H.L.); lg80204594@gmail.com (J.-G.C.); kmin2514@gmail.com (K.-M.K.); mjung@gnu.ac.kr (M.J.); scbaik@gnu.ac.kr (S.-C.B.); wklee@gnu.ac.kr (W.K.L.); kangssi@gnu.ac.kr (H.-L.K.); 2Department of Convergence Medical Sciences, Institute of Health Sciences, Gyeongsang National University, Jinju 52727, Korea; 3Department of Internal Medicine, Gyeongsang National University Hospital, College of Medicine, Gyeongsang National University, Jinju 52727, Korea; juny2278@naver.com (S.J.L.); dr202202@naver.com (Y.Y.J.); ljd8611@nate.com (J.D.L.)

**Keywords:** *Mycobacterium intracellulare*, molecular epidemiology, pulsed-field gel electrophoresis (PFGE), variable number tandem repeat (VNTR), mycobacterial interspersed repetitive units (MIRU), repetitive sequence based-PCR (rep-PCR)

## Abstract

*Mycobacterium intracellulare* is a leading cause of nontuberculous mycobacterial pulmonary disease, with a rapidly increasing prevalence worldwide. This bacterium, commonly distributed in soil and water, is known to be transmitted through the environment rather than between people. Therefore, it is imperative to establish distinguishable genotyping methods to understand the clinical outcome, disease relapses, and epidemiology. Therefore, in this study, representative band-based genotyping methods were performed using *M. intracellualre* clinical isolates, and their Hunter–Gaston discriminatory index (HGDI) was 0.947, 0.994, and 1 for variable number tandem repetition (VNTR), VNTR-mycobacterial interspersed repetitive units, pulsed field gel electrophoresis, and repetitive sequence based-PCR, respectively. Although VNTR showed relatively low HGDI, co-infection with other *M. intracellualre* strains could be determined by loci showing allele diversity from 0 to 0.69. Additionally, genetic distance of clinical isolates from Gyeongnam/Korea, and other regions/countries were visualized by minimum spanning tree (MST) using the globally available VNTR profiles. The results of MST revealed that *M. intracellulare* isolated from patients in Gyeongnam/Korea had specific VNTR genotypes, which may be evidence of the geographic distribution of *M. intracellulare* specific genotypes. The comparative results of genotyping techniques and geographical characteristics in this study may provide fundamental information for the epidemiology of *M. intracellulare*.

## 1. Introduction

*Mycobacterium intracellulare* belongs to the *Mycobacterium avium* complex (MAC), along with *M. avium*, and is the most common cause of nontuberculous mycobacteria-related pulmonary disease (NTM-PD) worldwide [1]. MAC infections can occur in both immunocompetent and immunosuppressed patients, *M. avium* is the most frequent organism in immunosuppressed patients such as HIV/AIDS infection or immunosuppressive therapy, and *M. intracellulare* may be responsible for about 40% of pulmonary infections even in immunocompetent patients [2]. MAC accounts for 60–80% of patients with NTM-related PD in the U.S. and Japan and 48–76% in Korea, and although the cases of *M. avium* have been increasing recently, *M. intracellulare* has been a main cause of MAC-PD [1,3,4]. The transmission route of MAC, including *M. intracellulare*, originates from the environment rather than between people [5]. This means that MAC infections occur mainly in plumbing and soil and are known to be easily exposed in residential environments [5,6]. In particular, in the case of environmental bacteria that are frequently exposed to individuals, it is important to understand the source of infection, relapse or reinfection, clinical epidemiology, and treatment outcome, and to support this, it is essential to develop a comparative genotype technology that can distinguish strains within the same species [7].

One of the most representative tools for epidemiological analysis is pulse field gel electrophoresis (PFGE), a technique that distinguishes individual strains, and this method is called the golden standard for finger printing [8,9,10]. However, new epidemiological tools have been developed to overcome the process complexity and time-consuming process of PFGE, and the representative of this is PCR-based technology, such as variable number tandem repeat (VNTR) and repetitive sequence based-PCR (rep-PCR). Ichikawa et al., 2010, and Dauchy et al., 2010, conducted a study to select effective VNTR loci for *M. intracellulare*, and applied the VNTR results using the selected VNTR loci to antibiotic sensitivity or clinical characteristics [11,12]. Rep-PCR is a method of amplifying between repetitive sequences scattered on the chromosome, which are preserved as BOX [13], ERIC [14], and REP [15] elements in various bacteria. The developed primers of rep-PCR are limited to those based on gram-negative bacteria, but clinical studies using rep-PCR have been frequently applied to epidemiological studies as they are one of the simplest tools for epidemiological analysis [5]. Rep-PCR has been used to investigate treatment outcomes and recurrence or reinfection of MAC-PD [5,16].

So far, several molecular genotyping methods have been applied to study many pathological or clinical and epidemiologic characteristics of MAC infection, and the key to the success of these studies is the development of a reliable and highly discriminating molecular genotyping method. However, since the PFGE, VNTR, and rep-PCR technologies developed and used so far may have different levels of discernment and clarity for different species, it is necessary to establish an effective molecular genotyping method for *M. intracellulare*. In the present study, we compared the PFGE, VNTR, VNTR-mycobacterial interspersed repetitive units (MIRU), and rep-PCR analyzes using *M. intracellulare* clinical isolates to calculate the discriminatory power and specificity. Moreover, minimum spanning tree (MST) analysis was performed using the VNTR genotyping results obtained in the previous literature and this study to analyze the epidemiological implication of Gyeongnam-derived *M. intracellulare*.

## 2. Materials and Methods

### 2.1. Bacterial Samples

A total of 101 *M. intracellulare* strains were used in this study, including two reference strains (*M. intracellulare* ATCC 13950 and *M. intracellulare* Asan 37128) and 99 clinical strains isolated from patients with PD at Gyeongsang National University Hospital from 2016 to 2018. Additionally, two strains of *M. intracellulare* that simultaneously infected a patient were included. First, the 19 *M. intracellulare* strains including *M. intracellulare* Asan 37128 were used for the comparative study of genotyping techniques (rep-PCR, PFGE, VNTR, and VNTR-MIRU), and VNTR and VNTR-MIRU were performed using all 99 clinical strains. Mycobacterial DNA was obtained using the AccuPrep^®^ Genomic DNA Extraction kit (Bioneer Inc.; Daejeon, Korea) following the manufacturer’s protocol.

### 2.2. VNTR and VNTR-MIRU

Sixteen pairs of VNTR primers [11] and seven pairs of VNTR-MIRU primers [12] were used for PCR amplification of VNTR and VNTR-MIRU. About 1 µL of each primer (10 pmol/each) and 5 µL of template were added to the Taq premix, and D.W. was added up to 20 µL. The PCR conditions are as follows: one cycle of 5 min at 95 °C; 35 cycles of 30 s at 95 °C, 1 min at 60 °C (58 °C for VNTR-MIRU), and 2 min at 72 °C; and one cycle of 7 min at 72 °C.

### 2.3. Rep-PCR

Primers (ERIC 1R (5′-ATGTAAGCTCCTGGGGATTCAC-3′), ERIC 2 (5′-AAGTAAGTGGGGGAGCG-3′)) previously described by Versalovic et al. (1991) were used for rep-PCR [17]. The PCR conditions were as follows: one cycle of 5 min at 95 °C, and 30 cycles of 30 s at 95 °C; one min at 40 °C; and 2 min at 72 °C, and one cycle of 7 min at 72 °C. The band pattern of the picture was recognized using the GelJ (version 2) program [18], which calculates the similarity between the band patterns of strains and draws the dendrogram of each strain (similarity coefficient, Dice; clustering method, Unweighted Pair Group Method with Arithmatic Mean (UPGMA)).

### 2.4. PFGE

PFGE was performed as a modification of a previously described method using an XbaI restriction enzyme [19]. Briefly, lysozyme and lysostaphin were treated with the bacterial culture to help cell wall digestion at a concentration of 100 mg/mL and 12.5 mg/mL, respectively. Bacterial suspension was mixed with 20 mg/mL Proteinase K and was followed by an equal volume of 1.3% low melt agarose. The agarose plugs were put in 1.5 mL of EC lysis buffer (6 mM Tris-HCl, 1M NaCl, 100 mM EDTA, 0.5% Brij-58 (NP-40), 0.2% Sodium deoxycholate, and 0.5% Sodium lauroylsarcosine pH 7.5) with 20 mg/mL Proteinase K for 24 h at 55 °C. The plugs were treated with 20 units of XbaI restriction enzyme (Takara Bio Inc., Shiga, Japan) for 4 h at 37 °C. The plugs were placed in each well of 2% Seakem ME agarose gel (Lonza, Rockland, ME, USA). Electrophoresis was performed in the TAE buffer using a CHEF Mapper (Bio-Rad Laboratories, Hercules, CA, USA). The agarose gel was analyzed using the GelJ (version 2) program (similarity coefficient, Dice; clustering method, UPGMA) [18].

### 2.5. Data Analysis

The Hunter–Gaston discriminatory index (HGDI, *D*) was used to calculate a value of ability that distinguishes individual strains to compare the effects of each genotyping method [20]. In addition, genetic diversity of VNTR was assessed by allelic diversity (*h*). Selander’s formula (*h*) was used to calculate the genetic variation of each locus used in the VNTR and VNTR-MIRU, and to compare it to the strains of the previous studies [21].

The minimum spanning tree was generated for visualization of the relationships among a large number of isolates. The first MST result was derived using the imsn function in the poppr package of the R program (version 3.5.3) [22] and the copy number of 16 loci of VNTR on 101 strains of this study and 266 strains of previous studies. The second MST result was also derived using copy numbers of 15 loci of VNTR and 6 loci of VNTR-MIRU by compiling 101 strains of this study and 47 strains of previous studies.

## 3. Results

### 3.1. VNTR and VNTR-MIRU Typing

VNTR typing for 16 VNTR loci and 7 VNTR-MIRU loci was performed using 19 *M. intracellulare* strains including *M. intracellulare* ATCC 1950 (Figure 1 and Table 1). Multiple bands representing multiple copy numbers were observed in loci VNTR10 and VNTR13 in sample number 28 from a patient with PD (Figure 1A). Since multiple bands of one sample were assumed to be the result of co-infection of two or more *M. intracellulare*, single isolation was carried out, resulting in two isolates representing each VNTR type (Figure 1B). These were used in this study as strains 28 and 29. Additionally, these strains of *M. intracellulare* that simultaneously infected a patient were included in the present study. The VNTR types were assigned according to the arrangement of copy numbers for each locus (Table 1). As a result, 19 strains were divided into 13 types from type A to M, with the most type D containing a total of four strains, whereas types G, H, and K had two strains each. Each of the remaining nine types had one strain each. The HGDI of the techniques was 0.947.

### 3.2. PFGE

As shown in Figure 2, PFGE was performed on 19 strains, and all strains were divided into four clusters. Except for the two strains corresponding to cluster I, the other strains were similarly assigned to clusters II, III, and IV. In particular, strains 25 and 27 (VNTR type K), 21 and 28 (VNTR type G), 22 and 29 (VNTR type H), and 14, 18, 30, and 20 (VNTR type D), showing the same VNTR types, were included in PFGE clusters II, III, and IV, respectively (Figure 2). In addition, two isolates 28 and 29 isolated from the same patient could also be separated by PFGE, and the HGDI of PFGE was 0.994.

### 3.3. Rep-PCR

As a result of the rep-PCR for 19 strains, the size range of the amplified bands was approximately 150 bp to 10 kbp, of which the bands ranging from 400 bp to 10 kbp were selected for analysis (Figure 3). There were about 10 bands per strain, and 19 strains were divided into 19 patterns depending on the pattern of the band, resulting in HGDI = 1. As shown in the dendrogram (Figure 3), 19 strains were largely divided into two clusters and three nonclustered strains, and strains 28 and 29 isolated from the same patient could be distinguished by rep-PCR.

### 3.4. Minimum Spanning Tree

MST analysis was performed using the VNTR and VNTR-MIRU data of clinical isolates from this study and previous reports from other countries or another region of Korea. Ninety-nine *M. intracellulare* clinical isolates from Gyeongnam, Korea, and two reference strains were compared with those from Japan (n = 226; [11,23,24]), Korea (n = 16; [23]), the United States (n = 15; [23]), and the Netherlands (n = 7; [23]), using the profiles of 15 VNTR loci (Figure 4; see also Appendix A). Compared to other countries, many strains derived from Japan were applied to this MST analysis; therefore, the strains from Japan were distributed for each branch of the MST analysis of *M. intracellulare*, but the strains from Gyeongnam, Korea, were also distributed overall (Figure 4). However, the cluster (green) containing the *M. intracellulare* ATCC 13950 seems to be a cluster with universal genotypes that include isolates from all countries and regions, but contains many strains derived from Gyeongnam, Korea, that show the same VNTR type (Figure 4). In addition, many Japanese-derived strains were included in the cluster (yellow), and the cluster (red) showed especially unique genetic characteristics of Gyeongnam, and all bacteria except isolated from the Netherlands consisted of only Gyeongnam-derived strains (Figure 4).

The second MST analysis was conducted to reconfirm the characteristics of the strains from Gyeongnam, Korea (n = 99), and the comparison with the strains from Seoul, Korea (n = 45; [25]), was conducted more precisely using 21 loci of VNTR and VNTR-MIRU excluding VNTR 11 and MIRU3 (Figure 5; see also Appendix A). As a result, the Gyeongnam-derived strains were found to form genetic clusters (red, green) different from those of Seoul (purple cluster) (Figure 5). Moreover, the reference strains were included in clusters related to Gyeongnam-derived strains.

### 3.5. Allelic Diversity of VNTR Loci

Comparing allelic diversity (*h*) on 101 Gyeongnam strains with the strains from previous studies, we checked the genetic variation of each VNTR locus between countries or other regions of Korea. As the *h* value is closer to zero, there is no genetic variation in the locus, and the closer it is to 1, the higher the variation. According to the VNTR and VNTR-MIRU loci, the *h* values of 0 to 0.69 were shown in Gyeongnam, Korea (Table 2). Also, the *h* values of VNTR 1, 6, 14, and 16 in Gyeongnam were 0. In addition, the *h* values of VNTR 5 and VNTR 8 were also 0.15 and 0.38 respectively, which is lower than that of other countries or regions in Korea (Table 2). 

## 4. Discussion

In the present study, VNTR, VNTR-MIRU, rep-PCR, and PFGE were performed to establish an effective genetic analysis method for *M. intracellulare* clinical isolates. The HGDI values for PFGE and rep-PCR were 0.994 and 1, and for the combination of VNTR and VNTR-MIRU, it was 0.947. These are typical band-based genotyping techniques, and their discriminative power is high. Although there are some differences in HGDI values according to each genotyping method, it is considered to be applicable to *M. intracellulare* in consideration of the accessibility of the experimental technique and interpretation.

VNTR and VNTR-MIRU are ways to explain the evolutionary relationship between closely related strains by changing the number of copies in the tandem repeat of each locus due to errors when DNA is transferred from bacteria to offspring [27,28]. This method for the *M. intracellulare* is based on the genetic sequences of the *M. intracellulare* ATCC 13950 reference strains [11,12]. The sequence of the PCR primers exhibits high specificity for the *M. intracellulare*, which can provide the ability to distinguish clinical isolates, and VNTR and VNTR-MIRU could be applied to the identification of co-infection. As shown in Figure 1A, the *M. intracellulare* co-infection was determined in VNTR10, VNTR 13, MIN19, and MIN22, which were highly discriminant loci with approximately 0.54–0.66 *h* values. However, interestingly, the Gyeongnam-derived strains showed an *h* values of 0 in VNTR1, 6, 14, and 16, and the Gyeongnam-derived strains are considered to have less genetic variation compared to other regions or other countries, as seen in Table 2. However, VNTR and VNTR-MIRU typing should perform 23 PCR sets for one strain, and the process of calculating the number of copies of each PCR band could be laborious. As shown in Table 2, there is various allelic diversity according to the loci of the VNTR; therefore, it seems to be able to use the selected loci with a high discrimination power.

Rep-PCR has been frequently used for the DNA fingerprinting of mycobacteria, including *M. tuberculosis*. In particular, the Diversilab Mycobacterium Kit (BioMerieux, Marcy l’Etoile, France) was developed as an automated commercial kit and was mainly applied to the evaluation of clinical outcomes or epidemiological studies of mycobacterial infections, but this kit is not currently available. Therefore, in the present study, ERIC primers were used among several primers that were used for rep-PCR. Basically, this method is based on a common sequence derived from *E. coli* and *Salmonella* Typhimurium, assuming that a false positive result can be obtained [17]. Moreover, although it is not clear whether this technique can be used to determine the distance and order of evolution in a strain [17], rep-PCR can be applied and used frequently for clinical diagnosis due to its simple and fast procedures, including differential results analysis through a band analysis program. As shown in the results of this study, the strain clusters formed by rep-PCR appeared to be unrelated to the clusters formed by PFGE and VNTR (Figure 3). However, as shown in Figure 2, since the strains of the same VNTR type are clustered together in the cluster by PFGE, the results of PFGE and VNTR genotyping may be related, and ultimately, the genetic variation is believed to be identified by PFGE and VNTR typing.

In order to understand the epidemiological implication of VNTR analysis, MST analysis was performed using the VNTR and VNTR-MIRU profiles of *M. intracellulare* provided in this study and previous studies. Gyeongnam-derived clinical isolates were clustered largely in two groups in the MST analysis, in which one cluster contained a large number of Gyeongnam-derived strains showing the same VNTR type, and also strains from Japan, the United States, the Netherlands, and Korea (Seoul), and *M. intracellulare* ATCC 13950 were clustered together (Figure 4). This meant sharing *M. intracellulare* represented the same or similar VNTR types with other regions and countries. In contrast, the other cluster showed a unique genetic characteristic of Gyeongnam, Korea, with the exception of the Netherlands-derived strain, in which all bacteria consisted of strains from Gyeongnam that were not related to any other region or country (Figure 4). This result is presumably the result of less genetic evolution of strains or relatively low influx of the strains with different genotypes from other regions.

However, MST does not mean the order of evolution by showing only the genetic distance, and it is not possible to infer the evolutionary history of the cluster, but it is applied to the study of geographic relevance. In the previous study of the relationship between the *M. avium* subsp. *hominissuis* (MAH) genotype and geographic origin, MAH from Japan seemed to be more closely related to the Korean lineage than Europe or the United States [23,29]. However, for *M. intracellulare*, there is still no reported relationship between geographic origin and genetic characteristics [12,23]. Therefore, the results of the present study could be new evidence that could reveal genetic relevance to the geographic origin of *M. intracellulare*. In addition, according to Fujita et al., 2013, the MAH strains isolated from the patient and the strains isolated from the soil sample in the area where the patient resides had the same VNTR type [24,30]. It has been reported that the VNTR type of MAH may be related to the progression of MAH-PD and the ability of MAH to infect human macrophages [31,32]. As such, it is expected that the VNTR and VNTR-MIRU results of the Gyeongnam-derived strains analyzed in this study can provide fundamental information for the study of the epidemiologic and clinical characteristics of *M. intracellulare*.

## 5. Conclusions

In the present study, the different characteristics of representative band-based genotyping techniques for *M. intracellulare* were clearly defined, and the discrimination power of each technique and the diversity of the VNTR and VNTR-MIRU loci were objectively determined by the HGDI and the allelic diversity, respectively. In conclusion, the band-based genotyping techniques including VNTR, VNTR-MIRU, rep-PCR, and PFGE have basically high discriminant power, and also different values guaranteed by each method, such as technical ease, cost, and genetic and evolutionary implications. Therefore, it is judged that these representative genotyping methods can be applied according to the value pursued in each study.

Furthermore, in the present study, the VNTR and VNTR-MIRU typing techniques were applied to the genotyping analysis for geographic and epidemiological studies on *M. intracellulare*, and we can confirm geographic characteristics representing unique genotypes of *M. intracellulare*. Therefore, the results of the present study are expected to provide fundamental information on the epidemiological study of *M. intracellulare* and to suggest a rational application of genotyping analysis to *M. intracellulare*.

## Figures and Tables

**Figure 1 microorganisms-08-01315-f001:**
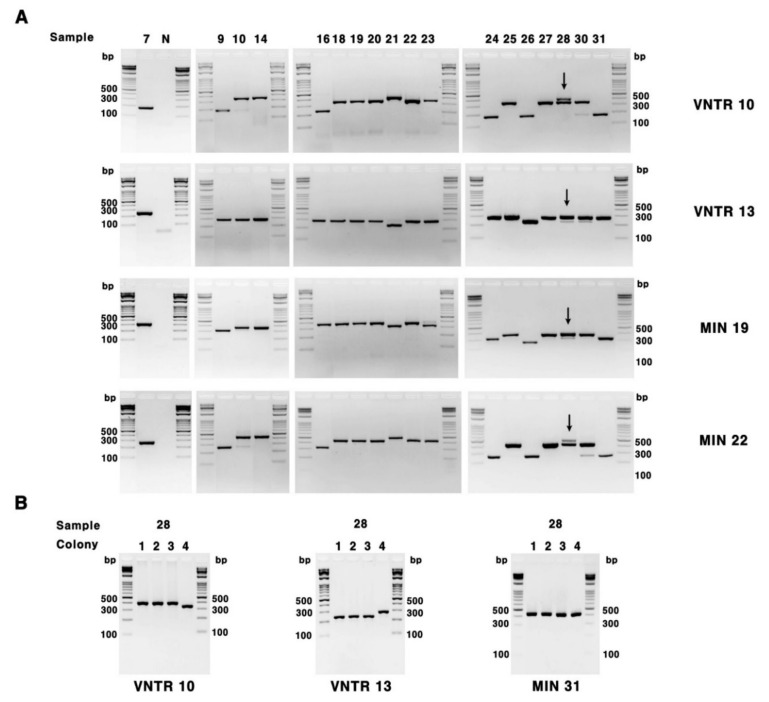
Representative results of variable number tandem repeat (VNTR) and VNTR-mycobacterial interspersed repetitive units (MIRU) using *M. intracellulare* isolates from patients with pulmonary diseases. (**A**) the amplicons of VNTR10, VNTR13, MIN19, and MIN22; (**B**) single isolation from no. 28 clinical isolate showing a multiband of VNTR. Arrows indicate samples showing multiple VNTR repeat bands suspected of co-infection with other *M. intracellulare* strains. Sample no. 28 was separated into two strains (28 and 29) through further single colony isolation; M, DNA ladder; N.C., negative control.

**Figure 2 microorganisms-08-01315-f002:**
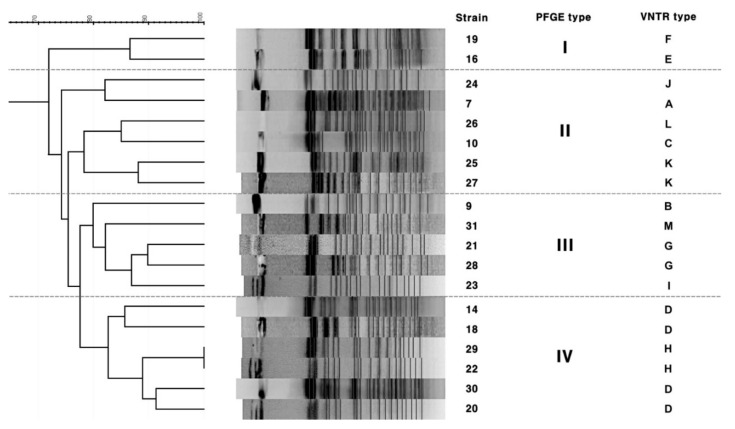
Dendrogram of the cluster analysis based on pulse field gel electrophoresis patterns of 19 *M. intracellulare* clinical isolates using XbaI.

**Figure 3 microorganisms-08-01315-f003:**
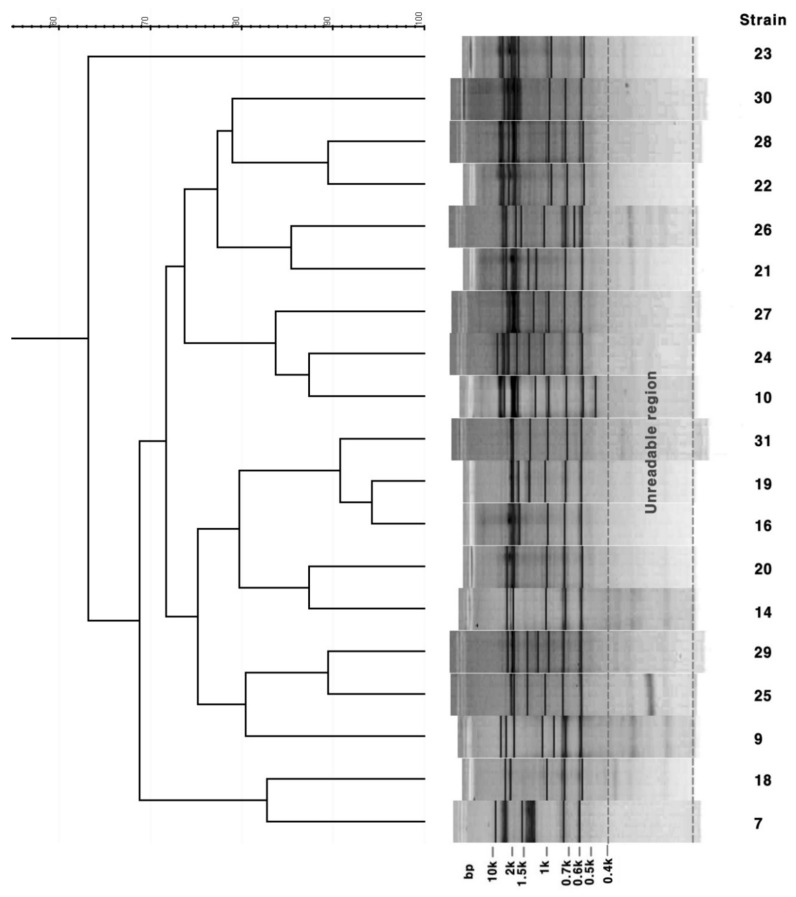
Cluster analysis of 19 *M. intracellulare* based on repetitive sequence based-PCR patterns. Bands below 400 bp were considered non-specific and unclear areas and were excluded from the analysis.

**Figure 4 microorganisms-08-01315-f004:**
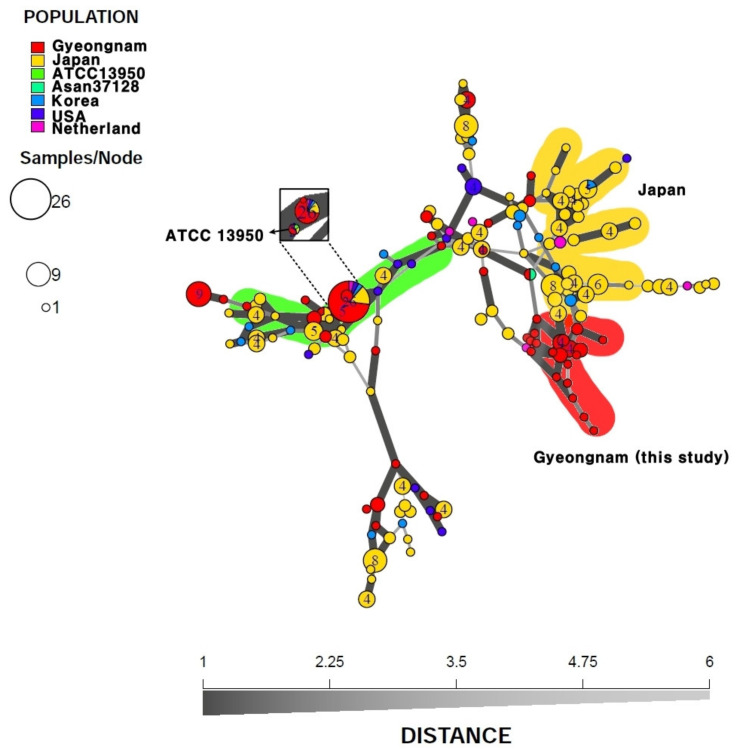
Minimum spanning tree of the variable number tandem repeat (VNTR) types for the 365 *M. intracellulare* strains. The minimum spanning tree was generated by R program using 15 VNTR loci, excluding VNTR 11 of previous studies and this study. Strains from Japan (n = 226), Korea (n = 16), the Netherlands (n = 7), the U.S. (n = 15), Gyeongnam (n = 101, this study), and reference strains (ATCC 13950, Asan 37128) were analyzed and their genetic distance was visualized.

**Figure 5 microorganisms-08-01315-f005:**
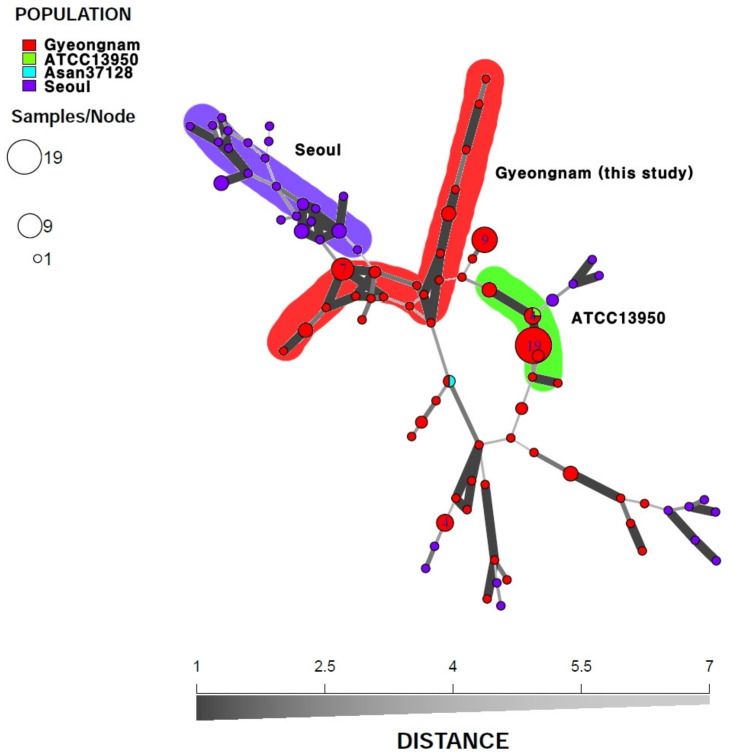
Minimum spanning tree of the variable number tandem repeat (VNTR) and VNTR-mycobacterial interspersed repetitive units (MIRU) types for the 146 *M. intracellulare* clinical strains from Korea. The minimum spanning tree was generated by R program using 21 VNTR and VNTR-MIRU loci, excluding VNTR 11 and MIRU 3 of previous studies and this study. The data of previous domestic studies (n = 45, Seoul) and the results from this study (n = 101, Gyeongnam) and reference strains (ATCC 13950 and Asan 37128) were analyzed in this study.

**Table 1 microorganisms-08-01315-t001:** Variable number tandem repeat (VNTR) and VNTR-mycobacterial interspersed repetitive units (MIRU) profiles of 19 *M. intracellulare* representative isolates.

loci Strains	VNTR																MIRU							Types ^1^
	1	2	3	4	5	6	7	8	9	10	11	12	13	14	15	16	3	18	19	20	22	31	33	
7 ^2^	2	1	1	2	2	3	2	2	1	1	4	2	2	2	2	2	2	2	2	2	0	1	3	A
9	3	1	1	4	2	3	2	0	1	1	4	2	3	2	1	2	2	2	2	2	0	2	5	B
10	2	1	1	3	2	3	3	2	2	4	2	2	3	2	2	2	2	3	3	3	3	2	5	C
14	2	3	1	3	2	3	3	2	2	4	2	2	3	2	2	2	1	3	3	1	3	2	4	D
16	2	2	0	4	0	3	3	2	0	2	2	2	3	2	1	2	3	1	3	1	1	1	4	E
18	2	3	1	3	2	3	3	2	2	4	2	2	3	2	2	2	1	3	3	1	3	2	4	D
19	2	2	0	4	0	3	3	2	0	4	2	2	3	2	1	2	3	1	3	1	3	1	4	F
20	2	3	1	3	2	3	3	2	2	4	2	2	3	2	2	2	1	3	3	1	3	2	4	D
21	2	3	2	1	2	3	2	2	2	5	2	2	2	2	2	2	3	3	2	4	4	2	3	G
22	2	3	2	1	2	3	3	2	2	4	2	2	3	2	2	2	3	3	3	4	3	2	3	H
23	2	1	1	3	2	3	2	2	2	4	2	2	3	2	2	2	1	3	2	2	3	2	4	I
24	3	1	1	4	2	3	2	0	1	1	4	3	3	2	1	2	2	2	2	2	0	2	3	J
25	2	2	1	1	0	3	3	2	0	4	2	2	3	2	1	2	3	1	3	1	3	1	4	K
26	2	1	1	1	2	3	1	2	1	1	5	2	2	2	1	2	2	2	1	2	0	1	3	L
27	2	2	1	1	0	3	3	2	0	4	2	2	3	2	1	2	3	1	3	1	3	1	4	K
28 ^3^	2	3	2	1	2	3	2	2	2	5	2	2	2	2	2	2	3	3	2	4	4	2	3	G
29 ^3^	2	3	2	1	2	3	3	2	2	4	2	2	3	2	2	2	3	3	3	4	3	2	3	H
30	2	3	1	3	2	3	3	2	2	4	2	2	3	2	2	2	1	3	3	1	3	2	4	D
31	2	3	0	2	2	3	2	2	1	1	4	2	3	2	2	2	1	2	2	2	0	2	2	M

^1^ VNTR types were assigned according to the order of arrangement of the copy numbers. ^2^
*M. intracellulare* Asan 37128. ^3^ It was two *M. intracellulare* isolated from a patient with pulmonary disease, and was determined to be co-infected.

**Table 2 microorganisms-08-01315-t002:** Allelic diversity of variable number tandem repeat (VNTR) and VNTR-mycobacterial interspersed repetitive units (MIRU) for strains used in previous studies and in this study.

	Allelic Diversity (*h*)							
loci	France (n = 52)	USA (n = 41)	Japan (n = 74)	Japan (n = 74)	Japan, Korea, The Netherlands, USA (n = 116)	Korea (n = 70)	Korea (n = 44)	Korea (n = 101)
VNTR 1	-	-	0.46	0.5	0.44	0.18	0.09	0
VNTR 2	-	-	0.58	0.73	0.58	0.69	0.5	0.61
VNTR 3	-	-	0.27	0.57	0.37	0.37	0.17	0.48
VNTR 4	-	-	0.69	0.72	0.71	0.5	0.61	0.61
VNTR 5	-	-	0.4	0.52	0.44	0.33	0.36	0.15
VNTR 6	-	-	0.65	0.57	0.63	0.21	0.53	0
VNTR 7	-	-	0.62	0.58	0.62	0.71	0.71	0.66
VNTR 8	-	-	0.52	0.6	0.55	0.55	0.51	0.38
VNTR 9	-	-	0.07	0.54	0.24	0.42	0.63	0.6
VNTR 10	-	-	0.56	0.68	0.57	0.27	0.46	0.6
VNTR 11	-	-	0.54	0.59	0.56	0.64	-	0.54
VNTR 12	-	-	0.3	0.39	0.31	0.39	0.44	0.3
VNTR 13	-	-	0.54	0.44	0.53	0.55	0.49	0.54
VNTR 14	-	-	0.21	0.45	0.25	0.27	0.13	0
VNTR 15	-	-	0.55	0.48	0.54	0.54	0.51	0.5
VNTR 16	-	-	0.09	0.04	0.11	0.21	0	0
MIRU 3	0.74	0.70	-	-	-	-	-	0.65
MIN 18	0.72	0.75	-	-	-	-	0.46	0.6
MIN 19	0.63	0.59	-	-	-	-	0.68	0.66
MIN 20	0.71	0.69	-	-	-	-	0.45	0.69
MIN 22	0.68	0.67	-	-	-	-	0.43	0.6
MIN 31	0.59	0.48	-	-	-	-	0.33	0.49
MIN 33	0.83	0.76	-	-	-	-	0.53	0.55
references	[12]	[7]	[11]	[24]	[23]	[26]	[25]	in this study

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
