# Peer review of "Comparative Evaluation of Band-Based Genotyping Methods for Mycobacterium intracellulare and Its Application for Epidemiological Analysis"

_microorganisms, 2020, doi:10.3390/microorganisms8091315_

Round 1

Reviewer 1 Report

The Manuscript titled, "Comparative evaluation of band-based genotyping methods for Mycobacterium intracellulare and its application for epidemiological analysis", is well written and presented. The study describes the representative band-based genotyping methods were performed using M. intracellualre clinical 24 isolates with a reasonable conclusion. This study seems to be in continuation with the earlier article, in Microorganism, January 2020 issue “Differential Genotyping of Mycobacterium avium Complex and Its Implications in Clinical and Environmental Epidemiology “.

I would have liked to see some representation of samples from African continent too. Any ways this is a good study, keep up the good work.

Author Response

We thank the reviewer for dedicating your valuable time to provide expert comments on our manuscript. As the reviewer’s comment, this manuscript is an extended study based on the “Differential Genotyping of Mycobacterium avium Complex and Its Implications in Clinical and Environmental Epidemiology “ published in the January 2020 issue of the journal “Microorganisms”. Thank you so much for reviewing the results of our expanded research.

Table 2 in this manuscript summarizes the allelic diversity of loci used in VNTR and VNTR-MIRU completed in the present and previous studies. We have tried to include the results of VNTR and MIRU's global research on Mycobacterium intracellulare as much as possible, but we are sorry that geographically global data were not collected because we could not find any data on the African continent. Thank you for review again.

Reviewer 2 Report

It is a well written manuscript dealing with important methodologies for the analysis of M.i strains. I think the authors should emphasize more the comparative aspect of the article - what is the most cost/effective methodology to be used?  are the methods equivalent in terms of results? - according to the data it appears yes.  Are there differences in costs or time to execute?

Author Response

We thank the reviewer for the excellent suggestions and comments, which are valuable for improving the quality of our manuscript. We fully agree with the reviewer’s comment that the comparative aspects of genotyping methods should be emphasized as the title of this manuscript is "Comparative evaluation of band-based genotyping methods for Mycobacterium intracellulare and its application for epidemiological analysis.".

In the present study, VNTR, VNTR-MIRU, rep-PCR and PFGE were performed to establish an effective genotyping method for M. intracellulare. As described in the Introduction section of this manuscript, one of the most representative tools for epidemiological analysis is PFGE (pulse field gel electrophoresis), a technology that distinguishes individual strains, and is said to be the gold standard for DNA fingerprinting. However, PFGE has significant disadvantages of process complexity and time consuming. To overcome this, new tools have been developed, and the representative ones are band-based technologies such as variable number tandem repeat (VNTR) and repeat sequence-based PCR (rep-PCR). PFGE is actually used as a standard to validate or compare these newly developed genotyping methods.

The developed primer of rep-PCR has a clear limitation that it can obtain false positive results with a primer based on gram-negative bacteria, but clinical research using rep-PCR is one of the simplest epidemiological analysis tools, so it has been often applied to epidemiological studies. Rep-PCR has been frequently used to study the clinical relevance and treatment outcome of mycobacteria, including M. tuberculosis.

VNTR-MIRU requires you to perform 23 PCR sets for one strain, and counting the number of copies of each PCR band can be challenging. However, in this study, since the locus of VNTR is very specific to the M. intracellulare, it was possible to determine that two or more bacteria of M. intracellulare were co-infected at the locus with high discriminative power. In addition, we can share the bacteria's VNTR profiles worldwide, and we were able to confirm certain advantages that can be used for epidemiological studies together using VNTR profiles.

The value of each genotyping technique, including opportunity cost and effectiveness, has been summarized, and the strategy of applying genotyping techniques to M. intracellulare was described in the Conclusion section of the manuscript. Newly added and organized contents in the manuscript are marked in blue. Thank you for your valuable advice again. (page 11-12, lines 280-294)
